# Stereotactic Radiation Therapy versus Brachytherapy: Relative Strengths of Two Highly Efficient Options for the Treatment of Localized Prostate Cancer

**DOI:** 10.3390/cancers14092226

**Published:** 2022-04-29

**Authors:** Manon Kissel, Gilles Créhange, Pierre Graff

**Affiliations:** Department of Radiation Oncology, Institut Curie, 26 Rue d’Ulm, 75005 Paris, France; manon.kissel@curie.fr (M.K.); gilles.crehange@curie.fr (G.C.)

**Keywords:** prostatic neoplasm, radiotherapy, brachytherapy, ultra-hypofractionated, stereotactic radiation therapy

## Abstract

**Simple Summary:**

Stereotactic radiation therapy consists of delivering ablative ultra-high doses of radiation to a precise target in a very limited number of fractions. Because of its ability to drastically shorten treatment duration and its potential radiobiological advantage, interest in this technique for the treatment of localized prostate cancer has increased over the past decade. At the same time, brachytherapy remains a time-tested technique that has excellent, proven outcomes with a very long follow-up, setting a standard of treatment for all subsets of the disease and a reference against which emerging techniques should be compared. We propose a critical literature review to report on the respective levels of evidence of stereotactic radiation therapy and brachytherapy for the treatment of localized prostate cancer.

**Abstract:**

Stereotactic body radiation therapy (SBRT) has become a valid option for the treatment of low- and intermediate-risk prostate cancer. In randomized trials, it was found not inferior to conventionally fractionated external beam radiation therapy (EBRT). It also compares favorably to brachytherapy (BT) even if level 1 evidence is lacking. However, BT remains a strong competitor, especially for young patients, as series with 10–15 years of median follow-up have proven its efficacy over time. SBRT will thus have to confirm its effectiveness over the long-term as well. SBRT has the advantage over BT of less acute urinary toxicity and, more hypothetically, less sexual impairment. Data are limited regarding SBRT for high-risk disease while BT, as a boost after EBRT, has demonstrated superiority against EBRT alone in randomized trials. However, patients should be informed of significant urinary toxicity. SBRT is under investigation in strategies of treatment intensification such as combination of EBRT plus SBRT boost or focal dose escalation to the tumor site within the prostate. Our goal was to examine respective levels of evidence of SBRT and BT for the treatment of localized prostate cancer in terms of oncologic outcomes, toxicity and quality of life, and to discuss strategies of treatment intensification.

## 1. Introduction

Brachytherapy (BT), which consists of implanting radioactive sources inside or near the tumor, was developed not long after the discovery of radium, more than a hundred years ago. Its use for the treatment of prostate cancer was at first delayed by the misconception that prostate adenocarcinoma was a radioresistant cancer [1]. The first prostate BT was described in 1911, with an intracavitary temporary radium implant throughout a urethral catheter [1,2,3]. The first transperineal implants were described in 1917, when Barringer inserted finger-guided radium needles [4]. In 1972, Whitmore described retropubic Iodine-125 seed implantation through an open operation [5]. In addition, with the development of ultrasound (US)-guided biopsies in the 1970s, techniques emerged for the precise transperineal placement of needles in the prostate guided by transrectal US-scanning [6]. In the 1980s, this allowed for the implantation of radioactive seeds into the gland under transverse and, later on, under longitudinal US-scanning. Concomitantly, developments in computer software rendered intraoperative dose-planning more efficient, allowing for live optimization of the implant. Since the 1990s, the technique has gathered growing interest.

At the same time, major technical improvements in EBRT have proven very popular in recent decades. The ability to spatially vary the intensity of an X-ray beam, together with the development of onboard imaging devices for image-guided radiation therapy, have allowed accurate dose delivery with a sharp dose gradient to avoid organs at risk (OARs) in close vicinity of the tumor. This led to the strategy of dose-escalated EBRT (DE-EBRT), which turned out to be beneficial for the treatment of localized prostate cancer, since numerous randomized trials comparing low dose EBRT (≤70 Gy) with DE-EBRT (≥74 Gy) clearly demonstrated a benefit of dose escalation with improved 10-year biochemical recurrence free survival (bRFS). This benefit did not translate into metastasis-free survival (MFS), cancer specific survival (CSS) or overall survival (OS), and late grade 2+ toxicity significantly increased [7]. Yet, there is now consensus that a total dose ranging between 74 Gy and 80 Gy (conventional fractionation of 1.8–2 Gy per fraction) is recommended, and recent data seem to indicate that even higher doses could further improve results [8]. Another approach to increase the therapeutic ratio relied on hypofractionation, which consists of delivering the treatment in fewer fractions with higher dose per fraction. Moderate hypofractionation (2.5–3 Gy per fraction) was the first to be tested successful in randomized trials and found not inferior to normofractionated regimens [9,10,11,12,13,14,15]. These results, added to the obvious benefit of shortening treatment duration in terms of patient comfort, healthcare resources optimization and costs limitation, have paved the way for ultra-hypofractionation (≥6 Gy per fraction). Ultra-hypofractionated EBRT was named stereotactic body radiation therapy (SBRT). Its development was made possible thanks to other technical evolutions including the use of micro-multileaf collimators, the ability to dynamically conform the X-ray field’s edges to the shape of a target while it rotates around the patient and the use of robotic arm-mounted linear accelerators. Excellent results in terms of local control and tolerance, initially seen in the treatment of small inoperable lung tumors and brain metastases, resulted in its broad acceptance in most radiation oncology units worldwide. SBRT has gained major interest for the treatment of prostate cancer over the last fifteen years and has become routine practice in many institutions.

With the emergence of SBRT, the proportion of patients treated for prostate cancer with EBRT continues to grow while BT has steadily decreased. From 2006 to 2016, among patients with a first-time diagnosis of prostate cancer in the US, the use of BT dropped by 17.0%. During this same period, SBRT increased by 1.6% even though most randomized trials evaluating ultra-hypofractionation were not yet published [16]. Several factors contribute to the loss of appeal of BT compared to SBRT. It is a time-consuming and invasive procedure that must be performed in an operating room under stringent aseptic conditions and it requires patient’s general or epidural anesthesia. The technique demands a high level of expertise, meaning that specific human resources must be dedicated and must be adequately trained through well-organized national training programs. Despite these constraints, prostate BT is recognized as a strongly efficient treatment with long-term follow-up and is considered an established reference against which new approaches should be measured.

The aim of this review was to report on the respective levels of evidence of SBRT and BT for the treatment of localized prostate cancer in terms of oncologic outcomes, toxicity and quality of life. Both techniques were compared head to head for the treatment of low-, intermediate- and high-risk disease. Strategies of treatment intensification for more aggressive forms of disease were discussed as well.

## 2. Radiobiology

### 2.1. Ultra-Hypofractionated EBRT

The dose-fractionation sensitivity of a tissue refers to its sensitivity to radiotherapy fraction size, in other words, to the dose delivered per fraction of treatment. It is quantified by a measuring unit named α/β. Tissues with a low α/β are relatively resistant to low doses and more sensitive to high doses per fraction. Indeed, in comparison with other tissues, prostate adenocarcinoma has a very low α/β ratio. That specificity makes it a perfect candidate for hypofractionation. The actual value of prostate cancer α/β ratio is still debated, historically considered as low as 1.5 to 2.5 Gy [17,18,19], but recent data suggest values ranging from 3 to 5 Gy [20]. In any case, the prostate α/β ratio is supposedly lower than that of its surrounding OARs (rectum and bladder), so the choice of an appropriate fractionation could result in an increased biologically effective dose delivered to the gland while still respecting dose limits to OARs.

The high dose per fraction used in SBRT drastically increases the biologically effective dose delivered to the prostate that is much higher than the nominal prescribed dose. The equivalent dose that the prostate would have received if it had been irradiated at 2 Gy per fraction (equivalent dose in 2 Gy per fraction-EQD2) can be calculated using the linear quadratic model. Based on a prostate α/β ratio of 1.5 Gy, the EQD2_1.5_ of the most common ultra-hypofractionated regimens (36.25 and 40 Gy in five fractions) would range between 90 Gy and 110 Gy. If a lower prostate α/β ratio is considered, for instance 3 Gy, the EQD2_3_ decreases between 74 Gy and 88 Gy. This illustrates the limits of radiobiological theorization and the need for a stringent clinical validation.

Ultra-hypofractionated regimens follow the classical 4Rs of radiobiology (Repair of sublethal radiation-induced cellular/DNA damage, Redistribution of cells within the cell cycle, Reoxygenation of the surviving cells and Repopulation of cells after radiation). These regimens create direct DNA damage (double-strand breaks or unrepaired/misrepaired breaks), but they are also believed to induce an indirect and delayed cellular death through the apoptosis of the tumor neovascularization endothelial cells [21,22,23]. It has been shown that a very high dose delivered in a single fraction causes a rapid wave of endothelial apoptosis with tumor cell death two to three days after. The pathway involves the translocation of acid sphingomyelinase (ASMase) from intracellular compartments to the plasma membrane, where it hydrolyses sphingomyelin, generating ceramide, a proapoptotic messenger. As ASMase is abundant in endothelial cells, those cells are particularly responsive to high dose per fraction. It seems that microvascular endothelial apoptosis is essential for tumor death because sublethal radiation-induced cancer-cell damage becomes lethal in the presence of endothelial apoptosis. Another added value of ultra-hypofractionation may reside in the massive tumor antigen presentation to immune system cells, thus creating a long-lasting antitumor immunity [22]. When a high dose per fraction schedule is applied, it generates local inflammation and apoptosis that recruit dendritic cells in the field of irradiation. Those cells adopt the tumor antigens and secondarily present these antigens to cytotoxic T lymphocytes providing a tumor specific immunity.

### 2.2. Brachytherapy

The basic 4Rs of radiobiology (Repair, Redistribution, Reoxygenation and Repopulation) apply to BT as well [24,25]. As fractionation is a key point for EBRT, dose rate is a major radiobiological parameter for BT. Distinction must be made between low-dose rate BT (LDR-BT) which corresponds to the permanent implantation of numerous radioactive seeds throughout the prostate that will slowly release radiation over several months and high-dose rate BT (HDR-BT) where a single high activity radiation source is moved between predefined positions inside applicators inserted within the prostate and releases radiation in a few minutes. It is striking that these two techniques have the same relative efficacy for the treatment of prostate cancer even though they are based on opposite rationales.

In both BT approaches, the dosimetric profile relies on a steep dose gradient from the radioactive sources located inside the prostate. This results in a massive heterogeneity of the distribution of dose with inevitable areas of ultra-high dose in some parts of the gland. These areas are believed to be partly responsible for the remarkable efficacy of BT. On the contrary, further away from the radioactive sources, the dose falls off quickly allowing simultaneous sublethal damage repair of normal tissues.

HDR-BT (>12 Gy/h) is perhaps the ultimate form of ultra-hypofractionation. It can be delivered in a single fraction or fractionated in a limited number of fractions. The effect on lethal DNA damage depends on the tissues’ repair capacity (α/β ratio) and, in the case of fractionated HDR-BT, on the dose delivered at each fraction and the interval between fractions. A minimum interval of 6 h is classically respected between fractions to let surrounding normal tissues repair their sublethal lesions. The biologically effective dose is estimated for HDR-BT using the same linear quadratic model than the one used for EBRT, keeping in mind its limitations for large doses per fraction. HDR-BT in some way resembles SBRT as it delivers treatment in a very limited number of fractions. Nevertheless, as aforementioned, HDR-BT is characterized by a massive heterogeneity in dose distribution inside the prostate [26]. Because the source is dynamically moved between chosen positions inside applicators, it is possible to optimize the dose distribution and favor ultra-high dose delivery to the area of the gland affected by the cancer. It has been hypothesized that SBRT dose distribution could also be optimized to mimic the intra-prostatic and peri-prostatic dose distribution of HDR-BT [27].

LDR-BT releases radiation at a very low dose rate (<0.4 Gy/h) over several months. The most common radioisotope for permanent implants is Iodine-125 (^125^I), but Palladium-103 (^103^Pd) and Cesium-131 (^131^Cs) have been used as well. The dose rate is dependent on the half-life of the radioisotope (59 days for ^125^I, 17 days for ^103^Pd and 10 days for ^131^Cs). In this continuous protracted mode of irradiation, sub-lethal damages become lethal in a cumulative dose effect (incomplete repair model) [28]. Biologically equivalent dose calculation in LDR-BT is very complex, since it needs to take into account the half-life of the radioisotope, the initial dose, the dose rate, the tumor’s doubling time and its repair capacity [29]. Yet, the extended duration of treatment has been hypothesized to be particularly promising for prostate cancer because of the long doubling time of prostate cancer cells [30]. Furthermore, ^125^I decays by electron capture to an excited state that decays immediately by gamma emission, with a maximum energy of 35 keV. With such very low photon energy, the relative biological effectiveness (RBE) of permanent implants is greater than other BT modalities [31].

Beyond the classic 4Rs of radiobiology, other effects of BT are yet to be determined. The effect on vasculature, the complex interactions with the tumor microenvironment and its immunogenicity are largely unknown. Yet, the unequalled sharp dose gradient may induce a high rate of tumor cell death followed by the release of tumor-specific antigens while minimizing unwanted dose to peripheral immune cells [24,25,32]. In that setting, BT could act as an “in situ” vaccination that increases the recruitment of immune cells in the tumor microenvironment [33].

Beyond pure radiobiological considerations, radiation effectiveness dramatically depends on spatial accuracy as well. While organ mobility is a concern in prostate SBRT, BT does not have to take prostate movement into account, as the radioactive sources are located inside the gland. Some SBRT dedicated accelerators such as Cyberknife ^®^ (Accuray Incorporated, Sunnyvale, CA, USA) guarantee a sub-millimeter delivery accuracy providing a sufficient intrafraction image-guidance frequency [34]. Even with real-time tracking techniques using fiducials implanted in the prostate, SBRT still requires the application of a margin around the gland to include set-up uncertainties. This results in the unavoidable irradiation of normal tissues.

## 3. Ultra-Hypofractionated Radiotherapy versus Brachytherapy in Low-Risk/Favorable Intermediate-Risk Prostate Cancer

Active surveillance is the option of choice for low-risk (LR) and selected favorable intermediate-risk (FIR) prostate cancers [35]. In most cases, it is a temporary option as it requires strict patient adherence with the risk of final rejection, and as a more significant tumor might be diagnosed over time. When an interventional curative treatment is deemed necessary, options for LR and IR disease include radical prostatectomy (RP), EBRT and BT. The European Association of Urology (EAU) [36] suggests that EBRT and BT offer similar outcomes and improved toxicity and quality of life over surgery (level 1 evidence) based on the results of the Prostate Testing for Cancer and Treatment Trial (PROTECT) and the Surgical Prostatectomy Versus Interstitial Radiation Intervention Trial (SPIRIT) [37,38,39]. BT has the advantage of brevity over EBRT, but the recent emergence of SBRT raises interest. Until now, no randomized trial has compared BT and SBRT head to head, so patient advice must rely on an indirect comparison of published prospective series regarding the two techniques.

### 3.1. Oncologic Outcomes

Oncologic outcomes after prostate LDR-BT can be assessed through the reporting of numerous large prospective cohorts. For instance, in a randomized comparison, Giberti et al. deemed LDR-BT and RP equivalent options for the treatment of LR disease, with a 5-year bRFS rate of 92% [40]. The single arm phase II RTOG 9805 study enrolled 101 LR prostate cancer patients treated with I^125^ monotherapy and reported 8-year cumulative incidences of biochemical and metastasis failure of only 8.0% and 1.1%, respectively [41]. J-POPS was a prospective multi-institutional registry of patients with predominantly LR and IR disease. For the 1792 patients who received BT monotherapy +/− androgen deprivation therapy (ADT), 5-year bRFS was 89.3% [42]. In a multi-institutional analysis of around 3000 prostate cancer patients treated with BT, including 960 IR disease, 8-year bRFS was 70% [43]. BT is a time-tested technique as series with very long follow-up have been published [44,45,46,47,48,49,50,51,52,53,54]. As summarized in Table 1, these mature results suggest a 10–15-year bRFS as good as 85–95% and 70–90% for LR and IR disease, respectively. A review published by the Prostate Cancer Results Study Group reported for LR patients receiving LDR-BT monotherapy, 10-year bRFS, distant metastasis rate, CSM and OS >86%, <10%, <5% and >85%, respectively. For IR patients, 10-year bRFS was >65% with most studies reporting a rate around 90% [55]. Data on the efficacy of BT for patients under 60 years old are available and reassuring, making it a valid option for younger patients as well [49,50,52]. For instance, with nearly 9 years of follow-up, a series of 600 patients <60 years old reported a 10-year bRFS of 95% and 90% for LR and IR disease, respectively [49]. Another series of 423 men <60 years old reported, with 10 years of follow-up, a 15-year bRFS of 88% [52].

Two randomized trials assessed the role of adding EBRT to LDR-BT for the treatment of prostate cancer patients predominantly belonging to the IR group (88%) [56,57]. The first trial (trial 44/20) randomized patients to either 44 Gy EBRT followed by a 90 Gy LDR-BT boost, or 20 Gy EBRT followed by a 115 Gy LDR-BT boost. The second trial (trial 20/0) randomized patients to either 20 Gy EBRT followed by a 115 Gy LDR-BT boost or 125 Gy LDR-BT monotherapy (no EBRT). Results were pooled in a secondary analysis after stratification between favorable (FIR) and unfavorable intermediate risk (UIR) groups [57]. Neither the addition nor the dose of supplemental EBRT influenced the outcome within each risk group. The RTOG 0232 was a randomized trial that compared LDR-BT monotherapy to pelvis EBRT plus LDR-BT boost specifically within the FIR group [58]. The addition of pelvis irradiation did not bring any oncologic advantage with a 5-year bRFS around 85% in both arms. Participants in the LDR-BT-monotherapy arm experienced fewer late events. It is therefore possible to conclude that EBRT should not be added to BT when treating LR and FIR patients.

The quality of the implantation of the radioactive sources is of paramount importance for LDR-BT efficacy. From a cohort of 639 patients with available post-implant dosimetric data, Zelefsky et al. demonstrated that a post-implant CT-based D90 (dose received by ≥90% of the prostate) >130 Gy for I^125^ and >115 Gy for Pd^103^ was associated with better 8-year bRFS [43].

HDR-BT monotherapy may be an alternative to LDR-BT monotherapy in selected patients with LR and FIR disease [59,60,61]. Level 2 evidence suggests that outcomes after HDR-BT and LDR-BT are similar in this population [62]. However, data regarding HDR-BT remain limited with shorter follow-up [36]. The technique seems associated with a 5-year bRFS ≥85% [59,60,61,63,64,65,66,67,68,69,70]. In a systematic review, grade 3–4 GI and GU toxicities were deemed <5% [71]. Various schemes of dose prescription are available: 34 Gy in four fractions of 8.5 Gy, 36–38 Gy in four fractions of about 9.25 Gy, 31.5 Gy in three fractions of 10.5 Gy and 26 Gy in two fractions of 13.5 Gy [72]. HDR-BT monotherapy delivered in a single dose should not be used as 19 Gy in 1 fraction was inferior to 27 Gy in two fractions in a randomized phase II trial [68].

Regarding SBRT, numerous prospective phase I and phase II studies are available (Table 2), and some have been pooled in systematic reviews and meta-analyses.

Despite notable heterogeneity of practices in terms of dose prescription, fractionation, target volume definition (seminal vesicles treated or not), dose objectives (target volume minimal coverage, acceptance of dose heterogeneity, organs at risk dose constraints) and technical conditions of treatment delivery (imaged-guided robotic linear accelerator or multileaf collimator linear accelerator equipped with on-board imaging), the reported oncologic outcomes were consistently excellent. The prescription dose varied from 32–38 Gy in four fractions to 33.5–50 Gy in five fractions, a frequent schedule being 36.25 Gy in five fractions of 7.25 Gy every other day. More extreme hypofractionation schemes have been reported such as 40 Gy in three fractions and 19 Gy or 24 Gy in one fraction but results are still very preliminary [99,100,101]. Single ultra-high dose schemes should be viewed with caution as they were tested inferior to fractionated schemes with HDR-BT. Jackson et al. pooled results of thirty-six prospective series comprising 6116 patients. Only 15% received ADT [102]. With a 3.2-year follow-up, the 5-year bRFS was 96.7% and 92.1% for LR and IR disease, respectively. Focusing on twelve series with long follow-up (median 7 years), Kishan et al. reported among 2142 patients (5% ADT) a 7-year bRFS of 87%, 87% and 74% for LR, FIR and UIR disease, respectively [103]. The most recent systematic review performed for the Hypofractionated Treatment Effects in the Clinic project (HyTEC) reported data from 1154 patients among 25 series [104]. The bRFS for a mixed population with no distinction between LR and IR disease was 90–100% and 94–100% at 2 and 5 years, respectively. Interestingly, by modeling the probability of 5-year biochemical freedom from recurrence, a dose/effect relationship was observed. Survival curves for LR and IR patients showed an upper-limit above which increasing the dose seemed needless. This was confirmed in a recent comparison of oncologic outcomes of patients receiving four different dose prescriptions of increasing intensity. The bRFS seemed better for a treatment intensification up to 40 Gy in five fractions but not greater [105]. On the contrary, for the small subgroup of HR patients included in the HyTEC analysis, the tumor control probability modeling seemed to leave more opportunity for oncologic outcomes improvement with greater dose escalation.

In addition to these numerous non-randomized prospective series, the HYPO-RT-PC trial (Ultra-hypofractionated versus Conventionally Fractionated Radiotherapy for Prostate Cancer) is a non-inferiority phase III trial in which 1180 localized prostate cancer patients were randomly assigned to ultra-hypofractionated EBRT (42.7 Gy in seven fractions of 6.1 Gy delivered every other day) versus normofractionated DE-EBRT (78 Gy in 39 fractions of 2 Gy every day) with no ADT [106]. IR disease was dominant (80%) with no distinction between favorable and unfavorable categories. The remaining 20% patients were from the HR category. With a 5-year median follow-up, the 5-year bRFS was 84% in both arms, and ultra-hypofractionated EBRT was deemed non-inferior to normofractionated EBRT (adjusted HR, 1.002; 95% CI: 0.758–1.325).

There is yet no direct prospective comparison of BT versus SBRT available in the literature. A monoinstitutional retrospective study recently published compared outcomes of a cohort of LR or IR patients treated between 2010 and 2018 either with LDR-BT for 219 patients (125 Gy) or SBRT for 118 patients (42.5 Gy/five fractions). With a 4.3-year median follow-up, 5-year bRFS was 91.6% versus 97.6% for LDR-BT and SBRT, respectively (*p* = 0.108) [107]. The National Cancer Database was used to compare survival outcomes of patients treated for FIR prostate cancer across different treatment modalities between 2004 and 2014 in the US [108]. There was no significant OS difference in pairwise comparisons of BT versus SBRT (HR, 0.804; 95% CI, 0.593–1.09; *p* = 0.16; 10-year OS, 67.02% vs. 64.2%) or SBRT versus DE-EBRT (HR, 1.096; 95% CI, 0.810–1.48; *p* = 0.55; 10-year OS, 64.2% vs. 70.9%). Men receiving BT had a small but statistically significant improvement in OS compared with those receiving DE-EBRT (HR, 0.881; 95% CI, 0.829–0.938; *p* < 0.001; 10-year OS, 69.8% vs. 66.1%). In a multicentric American retrospective study, PSA data were analyzed for 3502 men with LR (63.5%), FIR (24.8%), and UIR (11.7%) prostate cancers treated with SBRT (49.0%), HDR-BT (14.6%), or LDR-BT (36.4%) without upfront ADT from 1990 to 2017. bRFS was similar for all three modalities (*p* ≥ 0.27) [109]. A propensity score-matched analysis of a Canadian multi-institutional database was published in 2017, comparing, among other modalities, the outcomes of BT and SBRT with no ADT in a population of >600 LR prostate cancer patients with >5 years of follow-up. There was no significant difference in bRFS for SBRT versus LDR-BT [110].

Table 3 summarizes the numerous randomized prospective trials already published or still pending that evaluate the role of SBRT for the treatment of localized prostate cancer. These trials compare SBRT against BT or normofractionated DE-EBRT for the treatment of LR and IR disease. They also assess SBRT in a strategy of treatment intensification for more aggressive diseases as discussed later on in the manuscript.

The experience with BT has shown that the median time to biochemical recurrence is late (about 7 years) with about a quarter of relapses occurring after 10 years [47,54]. The publication of long-term results with SBRT will therefore be essential. In comparison with BT, the median follow-up of SBRT studies is relatively short (5–7 years). While pending more mature results, it seems reasonable to reserve SBRT for older patients (>70 years old for instance) and still favor BT for the treatment of younger ones when technically feasible. This has the additional advantage of proposing a non-invasive approach to the older patient. This age threshold is theoretical and numerous teams worldwide feel comfortable enough to propose SBRT regardless of patients’ age.

### 3.2. Genitourinary and Gastrointestinal Toxicity

The most common toxicity after SBRT and LDR-BT is genitourinary (GU) and is related to irritative bladder neck reaction (notably pollakisuria and urgency) and urethral obstruction (dysuria that can go as far as retention). Gastrointestinal (GI) toxicity is mainly related to a temporary increase in intestinal transit (diarrhea) and mild rectal bleeding.

Acute toxicity after prostate SBRT can be assessed through the preliminary results of the stereotactic arm of the PACE-B non-inferiority phase III trial [111]. A total of 845 prostate cancer patients, mainly classified as IR, were randomized between SBRT (36.25 Gy in five fractions) and normo or moderately accelerated EBRT (78 Gy in 39 fractions or 62 Gy in 20 fractions). During the 12 weeks post treatment, there was no difference in cumulative incidence of grade 2+ GU (27% vs. 23%) or GI (12% vs. 10%) toxicity [111]. The aforementioned HYPO-RT-PC trial reported 28% grade 2+ acute GU toxicity at the end of radiotherapy with weak evidence of an increased frequency in the SBRT group versus the normofractionated group (*p* = 0.057) [106]. Grade 2+ GI acute toxicity was below 8% in both arms with no statistical difference. In the meta-analysis published by Jackson et al., 32 studies reported prostate SBRT toxicity outcomes as cumulative incidence. The weighted rate of acute GU toxicity was 15.5% grade 2, 0.5% grade 3 and 0% grade 4. The weighted rate of acute GI toxicity was 6.1% grade 2, 0.06% grade 3 and 0.03% grade 4.

For LDR BT, rates of grade 2 and grade 3 acute GU toxicity have been reported around 40% and 10%, respectively [40,112,113]. Patients with a larger prostate volume, greater number of implanted needles and greater baseline International Prostate Symptom Score (IPSS) may have more acute toxicity [114]. The incidence of acute urinary retention requiring catheterization after implantation (grade 3) has been reported at 7–13%. [113,115]. Unlike BT [116], it appears that for SBRT the pre-existence of severe urinary disorders (IPPS ≥ 15) is not a limitation and that these disorders could even be improved after irradiation [117,118]. In total, the comparison of acute urinary toxicity profiles of BT and SBRT seems to favor SBRT [119].

Acute GI toxicity with LDR-BT is infrequent and is mainly represented by mild rectal bleeding. Grade 2 and grade 3 GI toxicity is in the range 1–8% and 0–1%, respectively [47,112,113,120,121,122]. In total, grade 2 acute GI toxicity could be slightly greater when treating with SBRT versus LDR-BT, but grade 3+ GI toxicity is almost non-existent whatever the technique used.

Regarding late GU toxicity after SBRT, the results of the PACE-B trial were recently updated. SBRT slightly increased the rate of grade 2+ GU toxicity observed 2 years after treatment in comparison with normofractionated EBRT (11.6% versus 5.5%, *p* = 0.003), mainly because of a temporary increase in the rate of grade 2+ frequency and urgency between 9 and 18 months post treatment [123]. The HYPO-RT-PC trial reported in the SBRT arm an estimated rate of late grade 2+ GU toxicity of approximately 5% at 2 and 5 years post treatment and cumulative 2- and 5-year incidence of late GU grade 2+ toxicity of 13% and 18%, respectively (with no significant difference with conventional fractionation) [106]. The meta-analysis by Jackson et al., with a median follow-up of 3.2 years, reported a weighted cumulative incidence of late GU toxicity of 12.1% for grade 2, 0.9% for grade 3 and 0.04% for grade 4 [102]. The systematic review by Kishan et al. reported a cumulative 7-year incidence of late GU toxicity of 12.3% for grade 2 and 2.4% for grade 3+. Grade 3+ complaints included urinary retention, hematuria, ureteral stenosis and frequency [103].

After BT, late grade 2+ GU toxicity varies widely from 2 to 41% but late grade 3+ GU toxicity is consistently low between 0 and 5% [47,112,113]. In the French cohort reported by Cosset et al., comprising more than 600 patients, the cumulative 10-year incidence of late grade 3+ GU toxicity was 6%. The annual rate was 2.5% in the first year and then gradually decreased to 0.2% at 5 years post treatment [47]. Urinary incontinence after BT is rare [112,116], much less frequent than after RP [37,40], and the risk dramatically increases if a prostate transurethral resection is performed after BT [46]. In the previously cited retrospective study comparing two cohorts of LDR-BT and SBRT patients, there was no significant difference in late grade 3 GU toxicity between the two techniques (0.9% vs. 2.5%, *p* = 0.238) [107].

Regarding late GI toxicity after SBRT, the HYPO-RT-PC trial reported an estimated rate of grade 2+ GI toxicity of approximately 1% at 2 and 5 years post treatment and a cumulative 2- and 5-year incidence of late grade 2+ GI toxicity of 6% and 10%, respectively (with no significant difference versus conventional fractionation) [106]. The meta-analysis by Jackson et al. reported, with a median follow-up of 3.2 years, a weighted cumulative incidence of late GI toxicity of 4.9% for grade 2, 0.4% for grade 3 and 0.08% for grade 4. The systematic review by Kishan et al. reported a 7-year cumulative incidence of late GI toxicity of 4.5% for grade 2 and 0.4% for grade 3+. Grade 3+ complaints were mainly rectal bleeding.

After LDR BT, rates of late GI toxicity are low as well, in the range, 0–10% for grade 2 and 0–2% for grade3+ [46,47,112,113,120,121,122]. A monocentric comparison of two cohorts of patients treated with LDR-BT or SBRT showed a lower rate of late grade 3 GI toxicity with LDR-BT (0.0% vs. 2.5%, *p* = 0.018), but these results are weak due to their retrospective nature [119]. In total, late GI toxicity appears to be low and tolerable with both BT and SBRT.

The HyTEC project provided a thorough analysis of published data that could help to correlate SBRT late toxicity (≥3 months) with the dose received by normal tissues [124]. Disparities between published studies in terms of toxicity measurement scales, time-points of assessment and cut-points chosen to define “clinically meaningful” side effects made a pooled analysis very challenging and the dosimetric conclusions quite weak. Nevertheless, it appeared reassuring that, when the dose constraints used in major clinical trials were respected, the overall incidence of severe toxicity remained very low (e.g., grade 3+ usually <3%). The most noticeable point was that high dose to the rectum was constantly associated with toxicity, so authors recommended to keep the rectum maximal dose to the prescription dose and to not exceed a prescription of 50 Gy in five fractions of 10 Gy (or equivalent). Actually, that dose level was the highest of a multi-center dose escalation study and resulted in 10% grade 3–4 rectal toxicity requiring temporary or permanent colostomy [125].

The identification of dosimetric factors predictive of LDR-BT toxicity has been inconsistent across studies. In a study of 712 patients treated with LDR-BT, both dosimetric and non-dosimetric factors were associated with late urinary toxicity. On multivariate analysis, the significant predictors for late grade 2+ GU toxicity were a high baseline IPSS, the extent to which IPSS increases post-BT, acute toxicity and high V150 (volume of prostate receiving ≥150% of the prescription dose) [114]. Prostate volume, number of needles implanted and brachytherapist’s experience were also found to be predictive of late GU toxicity [115,122,126].

In addition to physician-reported toxicity, patient-reported outcomes (PROs) collected via self-administered quality-of-life (QoL) questionnaires are of broad interest. An observational prospective study included 120 patients with LR and IR disease treated with LDR-BT monotherapy [127]. Patients were asked to answer to the Expanded Prostate cancer Index Composite (EPIC)-50 questionnaire at baseline and then regularly up to 10 years. Response rates dropped after 2 years but remained high enough to capture long-term consequences. On average, a deep and statistically significant deterioration was seen in the urinary summary score 6 weeks after implant and was rated severe decline for 65% of patients. The drop was consistent across all 4 domains: function, bother, incontinence, and irritative/obstructive symptoms. The score remained significantly impaired at 6 months but improved slowly to return to pretreatment level by 10 months. At 5 years, only 12% of patients still reported a severe decline in urinary QoL. At 10 years, a mild reduction in urinary function was noticed, with a small increase in urinary incontinence. A significant deterioration was seen in the bowel summary score 6 weeks after implant, but the drop was less important than the one observed in urinary QoL. It was rated severe for 36% of patients. The score returned to pretreatment level by 6 to 10 months and only 6% of patients reported severe decline in bowel QoL at 5 years. Sexual scores were low at baseline in the full cohort and dropped mildly 6 weeks post BT and never returned to pre-treatment levels. These decreases were not statistically significant.

Regarding SBRT, the prospective measurements of PROs from the HYPO-RT-PC trial were recently published [128]. At the end of irradiation, a clinically significant deterioration of scores related to bowel problems (stool frequency, rush to toilet, flatulence, bowel cramp, mucus, blood in stool, and limitation in daily activity) was more frequently observed in the ultra-hypofractionated EBRT arm than in the normofractionated EBRT arm. A major limitation of these results is that ultra-hypofractionation was delivered only using 3D conformal radiotherapy and not intensity-modulated radiotherapy (IMRT). These acute bowel symptoms remained moderate and disappeared over time with scores returning to baseline values after one year. There were no statistically significant differences in the proportions of patients with clinically relevant urinary and sexual functioning problems between the two treatment groups.

Several studies aimed to compare the difference of impact on QoL of BT and SBRT. A multi-institutional pooled cohort analysis of PROs (EPIC-26) measured before and post-treatment (up to 2 years) was conducted in a cohort of 803 localized prostate cancer patients treated with LDR-BT, IMRT or SBRT with no ADT. Regarding BT and SBRT, the mean EPIC domain scores declined from baseline for both techniques at 1–2 months post-treatment, and generally recovered at subsequent time-points, except for sexual function. Two years after BT, there were statistically significant declines from baseline for sexual (−23.6 points), bowel (−7.1 points), urinary incontinence (−6.3 points) and urinary irritation/obstruction domains (−6.1 points). Two years after SBRT, there were statistically significant declines for sexual (−14.0 points), bowel (−1.3 points) and urinary incontinence domains (3.4 points), though not for urinary obstructive domain (−0.2 points). Throughout the follow-up, there was significantly more BT than SBRT patients who developed changes from baseline that were classified as “clinically detectable” regarding urinary irritation, bowel symptoms and sexual symptoms. On multivariate analysis, BT had significantly worse change from baseline to 2 years for urinary irritation/obstruction and bowel domain score. There were no significant differences in sexual or urinary incontinence QOL among the treatments [129]. A mono-institutional study reported longitudinal PROs measurements after SBRT (112 patients), LDR-BT (342 patients) and HDR-BT (159 patients). IPSS worsened at all time points compared with baseline after LDR-BT and HDR-BT. At early/late time points, clinically significant IPSS variations were more frequent after LDR-BT than HDR-BT or SBRT. All modalities showed early and late Sexual Health Inventory for Men (SHIM) worsening with no temporal differences between techniques [130].

The already excellent tolerance of SBRT could be further improved with magnetic resonance-guided SBRT. A linear-accelerator or a cobalt-based radiation technology is equipped with an on-board MRI. This integration, known as MRI-guided RT, has been driven by the superior soft-tissue contrast, organ motion visualization, and ability to monitor tumor and tissue physiologic changes provided by MRI compared with conventional X-ray on-board imaging systems. The MRI can be used before daily treatment to allow online adaptation to anatomic changes between RT fractions and during treatment for targeting the prostate. This should allow better accuracy and better OARs sparing with no need for unpleasant insertion of intraprostatic fiducials [131]. Preliminary results are encouraging as no acute grade 3+ GU or GI toxicity and very limited impact on PROs have been reported [132,133,134]. Still, the clinical experience is very limited and more numerous and mature results are required to justify the inherent cost implications of the technique.

### 3.3. Erectile Dysfunction

Preservation of erectile function has long been considered a strong advantage of BT over RP and even EBRT. Langley et al. reported results from a cohort of <60 year-old patients treated with LDR-BT. Patient-reported symptom scores for erectile function were obtained using the five-item version of the International Index of Erectile Function (IIEF-5). Seventy-five percent of the patients who were potent before treatment kept their sexual potency 5 years after treatment [49]. Another study recorded before and after BT patient-reported erectile function using the Sexual Health Inventory for Men score (SHIM) and/or the Mount Sinai Erectile Function Score (MSEFS). A total of 64% (151/237) of patients potent before BT kept their erectile function with a median follow-up of 10.2 years [52] The two randomized studies comparing BT and RP favored BT regarding that topic. The SPIRIT trial suffered from poor accrual, making it difficult to derive a solid conclusion from this study, but with a median follow-up of 5 years, the ability to have an erection was preserved after BT for more than 60% of patients and was sufficient for intercourse in more than 50% of cases (sexual domain of the cancer-specific Expanded Prostate Cancer Index Composite—EPIC-26) [37]. In the randomized trial by Giberti et al., patients in the BT arm reported a significantly better erectile function than those in the RP arm at 6 months and 1 year post treatment using the IIEF-5 and EORTC-QLQ-PR25 questionnaires [40]. Therefore, BT should be offered to young patients who wish to preserve their sexual potency, given that it provides a biochemical control equivalent to RP or other modalities of radiation therapy.

Data on erectile toxicity after SBRT are scarce. In the HYPO-RT-PC trial, patients were asked by their physician if their erection could result in intercourse. The baseline rate of patients with adequate erections was 70% and fell to 35% at 5 years post treatment. Nevertheless, there was no significant difference with the decrease observed in the conventional fractionation group as well. Using the patient-reported Prostate Cancer Symptom Scale (PCSS) questionnaire, no difference was observed between SBRT and conventional fractionation [106]. Other studies reported rates of grade 2+ erectile dysfunction between 5 and 28% and grade 3 between 0 and 2% (Radiation Therapy Oncology Group criteria—RTOG) [73,82,84]. Dess et al. prospectively followed PROs regarding sexual potency after SBRT using the EPIC-26 questionnaire. Among a cohort of 184 patients with erectile function sufficient for intercourse at baseline, 57% kept that ability 2 years post SBRT and 45% at 5 years. Erection rates after SBRT were not statistically different from model-predicted rates after normofractionated EBRT or BT [135].

It is not possible to decide the superiority of one technique over the other given the heterogeneity of the populations included in the studies and the numerous confounding factors impacting erection regardless of the treatment technique (age, comorbidities, ADT use, baseline erectile dysfunction). However, SBRT seems to be a promising challenger to BT in that field. The TEMPOS trial (NCT03830788) carried out by the French GenitoUrinary group (GETUG) is a multicenter, randomized-controlled medico-economic study currently accruing that compares BT and SBRT for the treatment of LR and IR localized prostate cancers. Patients must be sexually potent at inclusion so that they can be prospectively assessed for emergence of erectile dysfunction. This study should provide interesting data in the coming years.

Few dosimetric data have been identified as predictive for sexual toxicity. The dose received by 50% of the proximal pillars of the corpora cavernosa (D50) could be predictive of erectile dysfunction after LDR-BT [136,137]. The volume of penile bulb receiving more than 35 Gy (V35) was associated with sexual quality of life after SBRT but inconstantly [138,139]. Desai et al. have initiated a functional anatomy SBRT trial (NCT03525262), to address sexual function outcomes after SBRT with dose restriction to the neurovascular bundles. This is a promising approach to determine whether sexual preservation can be accomplished with optimized SBRT.

### 3.4. Current Recommendations for Low Risk/Favorable Intermediate Risk PROSTATE Cancer

LDR-BT monotherapy is a validated option for the treatment of LR prostate cancer according to the National Comprehensive Cancer Network (NCCN), the European Association of Urology (EAU) and the American Brachytherapy Society (ABS) guidelines [35,140,141], but active surveillance should be encouraged in this category of patients [36].

NCCN guidelines also validate the use of LDR-BT monotherapy for the treatment of FIR disease, defined by the respect of all the following criteria [140]:ISUP grade ≤ 2;Plus one and only one feature among cT2b-cT2c or ISUP grade 2 or PSA 10–20 ng/mL;Plus ≤ 50% random biopsy cores involved with cancer;Plus none of the following features: ≥cT3a or PSA > 20 ng/mL.

EAU guidelines are slightly more restrictive limiting the indication of LDR-BT to FIR patients respecting all of the following criteria [35]:cT1b-T2a;Plus ISUP grade 1 with <50% of random biopsy cores involved with cancer or ISUP grade 2 with <33% of random biopsy cores involved with cancer;Plus PSA < 10 ng/mL.

ABS, NCCN and EAU guidelines recognize that fractionated HDR-BT monotherapy is a possible substitute for LDR-BT for the treatment of LR and FIR disease [35,140,142]. However, EAU guidelines state that patients should be informed that results are only available from limited series in very experienced centers. Single fraction HDR-BT monotherapy is not recommended.

ABS guidelines clarify some patient characteristics that should be considered before intending BT monotherapy [141]. Patients with large prostate volumes (>60 cc) should receive a careful assessment to verify that a minimal pubic arch interference is available to allow needle implantation. Patients with large median lobes (e.g., prostate protrusion into the bladder) may benefit from a limited TURP performed more than 4 months before implant. Patients with prior TURP defects may be candidates if there are sufficient prostatic margins (e.g., 1 cm) around the defect, and with careful attention to urethral-sparing dosimetry. Urodynamic testing can be helpful in patients with a baseline IPSS score ≥15–18. Optimizing urinary function with medications before implant is recommended. BT can be considered in patients with well controlled inflammatory bowel disease.

Regarding SBRT, the EAU guidelines still advise its use to be restricted to prospective clinical trials and to inform patients of the uncertainties of long-term outcomes [35]. On the contrary, the NCCN guidelines state that, for the treatment of LR and FIR cancer patients, SBRT alone (no ADT) is a validated option provided the use of appropriate technology and medical expertise [140].

## 4. Ultra-Hypofractionated Radiotherapy versus Brachytherapy in Unfavorable Intermediate-Risk/High-Risk Prostate Cancer

### 4.1. BT Monotherapy

As previously mentioned, trials 44/20 and 20/0 are two randomized trials evaluating the role of adding EBRT to BT in the treatment of patients with IR prostate cancer. After stratification between FIR and UIR, it appeared that although UIR patients had a greater incidence of biochemical failure, cancer specific mortality and overall mortality, neither the addition nor the dose of supplemental EBRT influenced outcomes within each risk group [57]. A multi-institutional study reported the incidence of distant metastases for IR prostate cancers treated with BT, RP or DE-EBRT. Treatment modalities seemed to provide similar 10-year incidence of distant metastases among UIR patients: 10.2% for BT, 11.6% for RP and 13.5% for DE-EBRT [143]. Hence, ABS states, with a weak strength of recommendation and moderate level of evidence, that LDR-BT monotherapy is an option for selected UIR disease (one single unfavorable intermediate risk factor) with organ-confined disease confirmed on mpMRI [141,142].

In contrast, HR disease has a substantial risk of extracapsular or seminal vesicles extension beyond the limits of the dosimetric profile of a typical LDR-BT implant. In that view, HDR-BT could bring an advantage, since the vector needles can be implanted outside the anatomic limits of the gland in specific areas where permanent implants would not be inserted because of the risk of major post-implant displacement (periprostatic fat and seminal vesicles). HDR-BT monotherapy has been deemed efficient for the treatment of selected HR disease, but more mature data are needed [59,144]. A large study comprised 718 patients with IR and HR disease treated with HDR-BT monotherapy (38 Gy/four fractions or 34.5 Gy/three fractions). ADT was given to 60% of HR patients. The 5-year biochemical control rate was high and identical between IR and HR patient (93%). The rates of late grade 3 GU and GI toxicity were 3.5% and 1.6%, respectively [60]. Another study reported poorer results. Among 111 HR patients treated with HDR-BT monotherapy (48 Gy/eight fractions, 54 Gy/nine fractions, or 45.5 Gy/seven fractions) and ADT (94%), bRFS was 81% and 77% at 5 and 8 years, respectively. The 5-year cumulative incidence of late grade 2–3 GU and GI toxicity was 5% and 4%, respectively [145]. A comprehensive literature review compared long-term oncologic outcomes of risk-stratified patients according to their treatment option. For HR patients, EBRT plus BT boost (+/− ADT) appeared superior to BT monotherapy, especially LDR-BT [55], such that BT monotherapy is not recommended today for that risk category [72,141,146]. This is also supported by the recent results of the POP-RT trial, which randomized HR patients treated with DE-EBRT plus ADT for prostate-only versus whole-pelvic irradiation [147]. Results validated the benefit of the prophylactic pelvic irradiation when the risk of cancer cell spread to the lymph nodes is high (>20%), as it is the case for most UIR and HR patients. It is then not surprising that prostate-only irradiation, regardless of the technique used, does not provide optimal locoregional control for the most aggressive disease. Potential benefit therefore lies in the combination of whole-pelvic irradiation plus a prostate-directed boost.

### 4.2. EBRT Plus BT Boost

Historically, the concept of BT boost was first described by Critz et al. among 1500 patients using LDR-BT with an open implantation technique (laparotomy). With a very strict definition of biochemical failure (PSA nadir greater than 0.2 ng/mL or a subsequent PSA increase above this level), the 10-year bRFS in the HR group was 61%. Despite non-use of ADT, this result was far better than that of EBRT for this risk group at that time [148].

Subsequently, a cohort of 104 patients with IR (40%) or HR (60%) prostate cancer and who were surgically staged node negative were randomized between EBRT alone (66 Gy/33 fractions) and EBRT (40 Gy/20 fractions) plus BT boost using a temporary iridium implant (35 Gy given in 48 h) [149]. Only the prostate was treated with no pelvic prophylactic irradiation. No ADT was given. With a median follow-up of 14 years, a significant improvement in biochemical control was reported in favor of the BT boost arm (HR 0.53, 95% CI: 0.31–0.88) but there was no significant difference in MFS, CSS and OS [150]. The lack of pelvic prophylactic irradiation, the no ADT policy and the low total dose of the EBRT alone arm make these results debatable.

The Androgen Suppression Combined with Elective Nodal and Dose Escalated Radiation Therapy trial (ASCENDE-RT) randomized 398 prostate cancer patients with IR (31%) and HR (69%) disease between a whole-pelvic EBRT at 46 Gy in 23 fractions followed by a prostate boost delivered either with EBRT at 32 Gy in 16 additional fractions (DE-EBRT standard arm) or by a LDR-BT at 115 Gy (LDR-BT boost experimental arm) [151]. All patients received 12-month ADT. With a median follow-up of 6.5 years, men randomized to the standard arm were twice as likely to experience biochemical failure defined by the Phoenix threshold of PSA nadir + 2 ng/mL (*p* = 0.004). The 5-, 7-, and 9-year b-RFS was 89%, 86%, and 83% for the experimental arm versus 84%, 75%, and 62% for the standard arm (log-rank *p* < 0.001), respectively. The LDR-BT boost benefited both IR and HR patients. If a more stringent definition of biochemical recurrence was used, such as the surgical PSA threshold of 0.2 ng/mL, the number of relapse events increased only in the standard arm. The 7-year bRFS after DE-EBRT significantly declined to 38%, while among the LDR-BT boost subset, it remained at 85% [152]. The results of the ASCENDE-RT trial were recently updated with a median follow-up of 10 years. The 10- and 15-year b-RFS was 85% and 80% for the experimental arm and 66% and 53% for the standard arm (*p* < 0.01), respectively [153]. However, no significant difference was observed for MFS, CSS and OS, but the study was not powered to detect differences between these survival endpoints. The ASCENDE-RT trial sets a standard as it compared LDR-BT boost to DE-EBRT (78 Gy) and included pelvic prophylactic irradiation. Of note, ADT duration was somewhat short for HR patients, but that point was similar in the two arms, and it did not seem to negatively impact the oncologic outcomes observed with LDR-BT boost.

The French GETUG-P05 study (NCT02271659) randomized 298 prostate cancer patients with IR and HR disease between a whole-pelvic EBRT at 46 Gy in 23 fractions followed by a prostate boost delivered with either EBRT at 34 Gy in 17 fractions (80 Gy total) or BT boost (110 Gy LDR or 14 Gy/1 fraction HDR). Results are pending.

HDR BT boost has been tested as well. In a systematic review of retrospective and non-randomized prospective studies assessing HDR-BT boost, the rate of freedom from biochemical recurrence at 5 years was 85–100%, 80–98% and 59–96% for LR, IR and HR disease, respectively. Among prospective studies with ≥5 years of follow-up, the actuarial 5-year bRFS was 90–92%, 83–87% and 63–69% for LR, IR and HR disease, respectively. For the whole population, the 5-year CSS, OS, local recurrence rate and distant metastasis rate were 99–100%, 85–100%, 0–8%, and 0–12%, respectively [154]. Another systematic review reported among 5000 patients, with a 10-year median follow-up, average bRFS of 95%, 91% and 82% for LR, IR and HR patients, respectively [155].

Several propensity score-matched analyses aimed to demonstrate the superiority of EBRT plus BT boost over DE-EBRT alone. Khor et al. among two matched cohorts of 344 patients each, with IR (60%) and HR (40%) disease, reported a 5-year bRFS of 79.8% and 70.9% for HDR-BT boost and DE-EBRT alone, respectively (*p* = 0.0011) [156]. Smith et al. among two matched cohorts of 194 IR patients each, reported a 5-year bRFS of 87.6% and 75.2% for HDR-BT boost and DE-EBRT alone, respectively (HR 2.08; *p* = 0.007) [157]. Similar results were observed when comparing a cohort of 127 patients treated with LDR-BT boost to a matched DE-EBRT cohort. The 5-year bRFS was 92.5% and 75.3%, respectively (HR 4.68; *p* = 0.001).

Hoskin et al. randomized 216 patients with LR (4%), IR (42%) or HR (54%) prostate cancer for EBRT alone (55 Gy/20 fractions) versus EBRT (35.75 Gy/13 fractions) plus HDR-BT boost (17 Gy/two fractions). Only the prostate was treated with no pelvic irradiation. ADT was given for 3 to 36 months depending on the risk group. Updated long-term results were recently published. With a median follow-up of 12 years, the median time to biochemical relapse was 137 months in the HDR-BT boost arm versus 82 months in the EBRT alone arm. The 6- and 12-year bRFS were 71% and 48% in the HDR-BT boost arm compared with 55% and 27% in the EBRT alone arm (log rank *p* = 0.008). There was no significant difference in MFS or OS. Once again, the lack of pelvic prophylactic irradiation and the low total dose of the EBRT alone arm weaken these results [158].

A UK multicentric non-randomized cohort study prospectively assessed oncologic outcomes of 812 IR (21%) and HR (79%) patients treated with a combination of EBRT and HDR-BT boost [159]. Patients could either receive whole-pelvic EBRT (46 Gy/23 fractions for 49%) or prostate-only EBRT (37.5 Gy/15 fractions for 51%) followed by HDR-BT boost (15 Gy/1 fraction) in both cases. With a median follow-up of 4.7 years, the 5-year bRFS was 89% in the whole-pelvic group versus 81% in the prostate-only group (*p* = 0.007). On a subset analysis, the benefit of whole-pelvic irradiation was maintained in the HR group only (84% versus 77%, *p* = 0.001).

The three aforementioned randomized trials comparing EBRT plus a BT boost versus EBRT alone reported significant improvements in bRFS in favor of the BT boost with a median follow-up of 10 to 14 years but failed to demonstrate a difference in MFS, CSS or OS. The very long-term biological efficacy of the combination of EBRT plus BT boost is reinforced by the results of numerous retrospective series as well. Regarding LDR-BT boost, studies reported a 12-year bRFS of 89–91% [160,161] and 16-year bRFS of 82% [162]. For HDR-BT boost, studies reported a 10-year bRFS of 54–84% [163,164,165,166]. Given the prognosis of HR disease, these data can be considered mature enough. The lack of MFS, CSS and OS benefit of adding a BT boost to EBRT may be partly due to the use of ADT at the time of recurrence. As reflected by a median time to castration resistant disease of 7 years after radiotherapy, some patients will be long-term responders to salvage ADT before dying of their cancer [167].

The role and the duration of ADT in UIR and HR prostate cancer patients undergoing extreme dose-escalated radiation therapy have remained unclear due to a lack of randomized evidence, with some non-randomized studies showing no benefit from ADT [168,169]. A recent publication using the 10-year data from the TROG 03.04 RADAR trial helped to resolve this uncertainty for HR patients [170]. In that four arms trial, patients with mainly HR (80%) prostate cancer were randomized to 6 or 18 months ADT with or without 18 months zoledronic acid [171]. The participating centers were asked to select their preferred dosing options from a predetermined range of doses. The dosing options by increasing order were 66, 70, and 74 Gy (2 Gy/fraction) and 46 Gy (2 Gy/fraction) followed by an HDR-BT boost (19.5 Gy/three fractions). Stratification ensured that the radiation dose was balanced across trial arms. Regarding distant progression, there was no evidence of an interaction between ADT duration and RT dose. After adjustment for RT dose, patients who received 18 months of ADT experienced a reduced risk of distant progression compared with men who received 6 months (HR 0.70, *p* = 0.002). In subgroup analysis, the HDR-BT boost group showed a significant benefit from longer duration ADT with a 40% relative reduction in distant progression (HR 0.61, *p* = 0.036). Regarding death attributable to prostate cancer, significant reduction in cancer-specific mortality was achieved with 18 months ADT versus 6 months (HR 0.70, *p* = 0.009) irrespective of RT dose. The reduction in all-cause mortality due to longer ADT duration did not reach significance [HR 0.84, *p* = 0.06]. Regarding IR patients, the recently presented results from the RTOG 0815 demonstrated a significant but modest benefit from the addition of 6 months ADT versus no ADT to extreme dose-escalated radiation therapy in terms of bRFS (5-year bRFS 14% versus 8%, *p* < 0.001), distant metastases (5-year rates 3.1% versus 0.6%, *p* < 0.001) and cancer-specific mortality (5-year rates 0.9% versus 0%, *p* = 0.007) [172]. In this trial, patients were treated with DE-EBRT alone (median 79.1 Gy) or EBRT plus BT boost (11%). OS was not improved, and the authors concluded that adding ADT to extreme dose-escalated radiation therapy for IR patients should be weighed against a modest benefit and an increase in acute adverse events relative to endocrine, sexual and metabolic functions translating into a temporary (18 months) worsening of the corresponding PROs.

The combination of EBRT plus BT boost seems to stay relevant even when it is compared with very high dose EBRT. In a retrospective study comparing patients treated with IMRT up to 86.4 Gy to the whole prostate and a combination of IMRT (up to 50.4 Gy) plus a BT boost, the BT boost method resulted in better bRFS and MFS [173]. However, the use of the simultaneous integrated boost technique (SIB) could be a more effective approach. With SIB, a differential dose per fraction is delivered to selected sub-regions during the same treatment session, resulting in different total nominal doses given to different target volumes in the same number of fractions. This allows for drastically increasing the dose delivered to the area of the prostate gland where the index lesion has been located on biopsy mapping and multiparametric magnetic resonance imaging (mpMRI). The phase III randomized FLAME trial recently validated the benefit of the SIB focal boost technique in a population of 571 patients (84% HR) treated with EBRT (prostate only) plus ADT with or without SIB [8]. In the SIB arm, a total dose of 95 Gy was prescribed to the index lesion. With a median follow up of 6 years, the 5-year bRFS was 92% in the SIB arm and 85% in the no SIB arm (*p* < 0.001). The tolerance of the focal boost was excellent with a 5-year cumulative incidence of grade 3+ GU and GI toxicity of 5.6% and 1.4%, respectively. The toxicity profile in the FLAME trial seemed to be more favorable than the one reported with EBRT plus BT boost even though both modalities have not yet been compared upfront.

The combination of EBRT plus BT boost and upfront ADT seems especially relevant for the treatment of very aggressive tumors with the highest Gleason scores. The retrospective analysis of a large monocentric cohort of 487 patients with biopsy Gleason score 9–10 prostate cancer treated with either EBRT (median dose 76.4 Gy) or EBRT plus BT boost (LDR or HDR technique) or RP reported best results when using the BT boost approach [174]. Patients treated with radiation therapy received upfront ADT for a median duration of 24 months with EBRT and 8 months with EBRT plus BT boost. The 5- and 10-year DMFS were significantly higher with EBRT plus BT boost (94.6% and 89.8%) than with EBRT (78.7% and 66.7%, *p* = 0.0005) or RP (79.1% and 61.5%, *p* < 0.0001). These results were confirmed in a larger cohort of 1809 patients from 12 tertiary centers [175]. In multivariate analysis, EBRT plus BT boost was associated with a significantly lower rate of distant metastasis (HR of 0.27 [95% CI, 0.17–0.43] for RP and 0.30 [95% CI, 0.19–0.47] for EBRT) and a significantly lower prostate cancer-specific mortality (HRs of 0.38 [95% CI, 0.21–0.68] for RP and 0.41 [95% CI, 0.24–0.71] for EBRT). Within the first 7.5 years of follow-up, EBRT plus BT boost was associated with significantly lower all-cause mortality (HRs of 0.66 [95% CI, 0.46–0.96] for RP and 0.61 [95% CI, 0.45–0.84] for EBRT). After the first 7.5 years, the differences were not significant anymore.

### 4.3. Focus on the Toxicity of the Combination of EBRT Plus BT Boost

The encouraging oncologic outcomes of the ASCENDE-RT trial have been clouded by an unusually high rate of severe toxicity [176]. In comparison with DE-EBRT, the LDR-BT boost increased the risk of acute grade 2 GU toxicity (30.0% versus 15.8%, *p* < 0.0001) and was responsible for five acute grade 3 GU events (urinary frequency more often than hourly, severe dysuria, urge incontinence, bladder drainage and intermittent self-catheterization) when only 1 event was observed in the DE-EBRT arm (lumbar plexopathy and neurogenic bladder, thought to be secondary to Zoster reactivation). No acute grade 4+ GU events were recorded in either arm. The acute GI morbidity did not differ between the arms and was limited to grade 0–2 events. The rate of severe late GU toxicity was significantly increased as well, with a worrying 5-year cumulative incidence of grade 3 GU events of 18.4% for LDR-BT boost versus 5.2% for DE-EBRT (*p* < 0.001). Half of the late severe GU toxicities observed in the LDR-BT boost arm were urethral strictures and many involved the membranous urethra requiring dilatation. Other severe toxicities included dysuria needing transurethral prostate resection, severe urinary incontinence, urinary frequency >1x/hour, radiation cystitis requiring hyperbaric oxygen. In addition, 1 individual in each treatment arm developed grade 4 hematuria requiring transfusion/hospitalization. Of note, because many of the late grade 3 GU events resolved with treatment, the prevalence was less than half the cumulative incidence reaching 8.6% at 5 years in the LDR-BT boost arm versus 2.2% in the DE-EBRT arm (*p* = 0.058). There was also a non-significant trend for worse late GI morbidity in the LDR-boost arm (5-year cumulative incidence grade 3 GI events 8.1% versus 3.2%, *p* = 0.124).

The ASCENDE-RT team acknowledged that there were major flaws in the technique used for the implantation of the radioactive seeds that could explain the unexpected high level of urethral strictures. Notably, the protocol specified a too generous inferior margin in defining the BT target volume because of uncertainty regarding the location of the prostatic apex on previous generation transrectal ultrasound imaging. It was then stated that technological advances may reduce adverse events while maintaining the gain in bRFS. Other authors had already demonstrated that improving the quality of the implant could drastically influence BT toxicity [177].

HDR BT boost has also been historically associated with a substantial risk of urethral stricture, up to 8% at 2 years [178]. Nevertheless, prior to the era of transrectal ultrasound guidance, the implant needle insertion depth was guided solely by the surrogate of the needle length protruding from the perineal plate. This tended to systematically degrade the coverage of the base while overdosing the apex [179].

In Hoskin’s trial, using modern HDR-BT technique, the actuarial incidence of severe GU or GI toxicity did not significantly differ between the HDR-BT boost and the EBRT alone arms. Notably, regarding urethral strictures needing surgical management, the actuarial incidence rates at 6 and 12 years were 6% and 10% with HDR-BT boost versus 3% and 8% with EBRT alone (*p* = 0.3) [180]. The RTOG 0321, one of the first phase II studies evaluating the concept of HDR-BT boost (19 Gy/two fractions after 45 Gy EBRT), reported reassuring rates of severe toxicity. With a median follow-up of 2.5 years, the rate of grade 3–4 toxic effects was 2.6%, with a rate of urinary stricture of only 0.7% [181]. Other modern series have reported a favorable toxicity profile with HDR-BT boost. The Sunnybrook hospital prospectively reported, with a median follow-up of 5.2 years, an incidence of late grade 3+ GU toxicity of 4% and late grade 3+ GI toxicity of 0% [182]. In a large monocentric retrospective series, Spratt et al. reported 7-year actuarial rates for late grade 3 GU and GI toxicities of 1.4% and 1.4%, respectively [173]. In a comprehensive systematic review of retrospective and non-randomized prospective studies assessing HDR-BT boost, less than 6% of late grade 3–4 GU and GI toxicities have been reported, though most of the toxicities were due to urethral stricture. When focusing on prospective series with ≥4 year median follow-up, the rates of grade 3–4 GU and GI toxicities were 0–12% an 0–8%, respectively [154].

Besides dosimetric quality constraints and interoperator variability, predicting factors for late toxicity include initial presence of symptoms [183], ADT use, older age (>~65 years) [184], previous transurethral resection of the prostate [185] dose per HDR fraction [185], high blood pressure [185] and diabetes [186].

The addition of a BT boost seems to have a limited impact on erectile dysfunction in comparison with EBRT alone. In the ASCENDE-RT trial, among men reporting adequate baseline erections, 45% of LDR-BT boost patients reported similar erectile function at 5 years versus 37% after DE-EBRT (*p* = 0.30) [176]. Similar rates of potency alteration were reported in prospective series testing the HDR-BT boost. Among patients with normal erectile function at baseline, 41% maintained their normal function, 26% developed moderate impaired potency and 32% developed severe erectile dysfunction at 5 years [182]. Some authors suggested that a vessel-sparing approach may be relevant to preserve erectile function in the EBRT plus BT combination setting [187,188].

In terms of quality of life, studies seem to converge towards a lower quality of life for patients treated with EBRT plus BT boost versus EBRT or BT alone. In the RTOG 0232 randomized controlled trial comparing EBRT plus BT to BT alone, significant differences favoring the BT alone arm were reported at 2 years in the urinary, bowel, and sexual domains. However, none of these differences were deemed clinically significant [189]. The Comparative Effectiveness Analyses of Surgery and Radiation in localized prostate cancer (CEASAR) study is an observational study designed to compare the effectiveness of different treatments for localized prostate cancer. It employs a prospective cohort study design, using tumor registries as cohort inception tools. The primary outcome is quality of life assessment. The study included 578 men treated with EBRT and 109 treated with EBRT and LDR-BT boost. On multivariable analyses, men receiving LDR-BT boost reported worse urinary irritative function at 6 months (*p* < 0.001), 12 months (*p* < 0.001), and 36 months (*p* = 0.034) and worse bowel function at 12 months (*p* = 0.002) but not after. There was no significant difference in sexual function between treatment groups [190]. A clinically significant decline in mean scores was noted in both arms of the ASCENDE-RT trial for role physical and sexual function scales, but there was a significantly larger drop in mean scores in the LDR-BT boost group for physical function (*p* = 0.03) and urinary function (*p* = 0.04) [191].

### 4.4. SBRT Monotherapy

While SBRT has proven its efficacy for the treatment of LR and IR patients, data are weaker regarding the treatment of HR patients. The very limited number of HR patients (in most cases fewer than 50 patients) in published prospective SBRT series (Table 2) poses a major risk of selection bias. A recent attempt at a systematic review of all retrospective and prospective series about the treatment of HR patients with SBRT monotherapy highlighted the paucity of data available [192]. Biochemical control rates ranged from 82% to 100% after 2 years and 56% to 100% after 3 years. Only rare series reported data with longer follow-up. The prospective series with the longest follow-up (7 years) included only 38 HR patients with a 7-year bRFS of 65.0%. Results were not much better for 47 UIR patients, with a 7-year bRFS of 68.2% [92]. The only randomized data available are those from the aforementioned HYPO-RT-PC trial, which included only 11% HR patients. However, the results have been presented for the entire population with no sub-analysis by risk groups [106]. More recently, the Stereotactic Body Radiotherapy for High-Risk Localized Carcinoma (SHARP) consortium collected individual data of 344 HR patients prospectively treated with SBRT in 7 institutional phase II trials and prospective registries [193]. Only patients with a minimum follow-up of 2 years were included but the resulting median follow-up for the entire cohort remained low (4.1 years). Patients received five fractions of 7 Gy, 7.5 Gy or 8 Gy. ADT was given to 72% of patients with a median duration of 9 months and only 19% received an elective nodal irradiation. Estimated 4-year bRFS and DMFS rates were 81.7% and 89.1%, respectively. The crude incidences of late grade 3+ GU and GI toxicity were 2.3% and 0.9%, respectively. Lastly, a national cancer database analysis was performed to assess the oncologic outcomes of UIR and HR patients treated with external radiation (BT excluded) and ADT with curative intent in the US between 2004 and 2016. The aim was to compare efficacy of SBRT plus ADT versus conventional EBRT (normofractionation or moderate hypofractionation) plus ADT in that setting of patients. A total of 558 patients treated with SBRT (defined as five fractions of ≥5 Gy per fraction) and 40.797 patients treated with conventional EBRT (defined as ≤3 Gy per fraction and a total dose ≥60 Gy) were included. With a median follow up of 6 years, on multivariate analysis accounting for age, race, and comorbidity, there was no difference in the estimated 6-year OS between men treated with SBRT versus conventional EBRT [194]. The authors concluded that these results support recent NCCN guideline updates, which validate SBRT as an option for higher risk patients [140]. However, these encouraging results should still be interpreted with caution as UIR and HR groups have been recently identified as predictors for post-treatment biopsy outcomes after prostate SBRT [195]. A total of 257 prostate cancer patients treated with SBRT (32.5 Gy to 40 Gy in five to six fractions) +/− ADT (26.5%) underwent a post-treatment biopsy performed approximately two years after treatment to evaluate local control status. Patients belonged to LR (17%), FIR (33%), UIR (40%) and HR (10%) groups. In univariate analysis absence of ADT, low prescription dose (<40 Gy versus ≥40 Gy) and aggressive disease (UIR-HR versus LR-FIR) were associated with an increased risk of post-treatment positive biopsies. In multivariate analysis, only low prescription dose (HR 2.75; *p* = 0.008) and aggressive disease (HR 2.34; *p* = 0.026) remained predictive. A positive biopsy was associated with a significantly higher likelihood of subsequent PSA relapse at 5 years. A significant interaction was found where the association between dose and positive biopsy was stronger for those with UIR-HR disease (OR: 7.25) compared to those with LR-FIR disease (OR: 1.45) with *p* = 0.027. This is in agreement with the assumption that UIR and HR cancers could better benefit from dose escalation than the more favorable group risks.

When treating HR patients, the use of SBRT targeting the prostate gland only raises concern about disregarding potential microscopic spread of the disease in regional pelvic lymph nodes. To overcome that risk, elective nodal ultra-hypofractionated radiation therapy has been tested. Initial experience led to caution since the FASTR trial had to be prematurely discontinued owing to an unacceptable rate of acute and late toxicities notably a high rate of severe late GI toxicity [196]. In that phase I/II trial, 15 HR patients were treated with 12 months of ADT and radiation therapy was delivered using 25 Gy to pelvic nodes delivered synchronously with 40 Gy to the prostate given as 1 fraction per week. Three patients (18%) experienced late grade 3 GI toxicity (rectal bleeding requiring argon laser coagulation) and 1 patient (6%) experienced late grade 4 GI toxicity (bowel obstruction requiring a partial colectomy). Yet, much more reassuring results have been reported by the Sunnybrook Hospital which conducted four prospective phase II trials (SATURN, SPARE, 5STAR, 5STAR-PC) recently merged in a pooled analysis [197]. A total of 165 patients were prescribed a pelvic nodal irradiation at 25 Gy in five weekly fractions while the prostate was simultaneously treated at 40 Gy in five fractions (+/− a SIB boost up to 50 Gy in five fractions to the mpMRI dominant lesion) or at 25 Gy in five fractions followed by an HDR-BT boost (15 Gy single-fraction). ADT was given to 85% of patients for a duration of 6–18 months depending on the risk-group classification. The worst physician reported acute GU and GI toxicity was 48% and 7.5% for grade 2, and 2.7% and 0% for grade 3 (no grade 4–5). With a median follow-up of 38 months, the cumulative incidence of late grade 2+ GU toxicity was 41.1% with 1.5% grade 3 and no grade 4–5. For late grade 2+ GI toxicity, it was 10.5% with no grade 3–5. With favorable early oncologic outcome (98% 3-year bRFS), this approach was deemed worth being tested in phase III trials.

The addition of ADT to conventional EBRT versus EBRT alone for the treatment of UIR and HR patients has been widely proven beneficial in terms of bRFS, MFS and CSS [198,199,200,201,202], while FIR patients are candidates for EBRT alone [143,203]. The benefit of adding ADT seemed confirmed even in the field of DE-EBRT (≥74 Gy) even though no significant difference in OS was observed [172,204,205,206]. For IR patients, there is no obvious benefit of prolonging neoadjuvant/concomitant ADT beyond a total duration of 4–6 months [207,208]. On the other hand, HR patients take advantage of a prolonged adjuvant ADT for a total duration of 24–36 months with the option of reducing to 18 months [209,210]. Dose escalation with EBRT does not decrease the need for long term ADT [170,171,211]. However, recent data suggest that HR patients treated with a combination of EBRT plus BT boost are those who would most benefit from a reduction in ADT to 18 months and maybe shorter, whereas others should receive a longer duration of ADT [212].

Among series that reported on the use of SBRT for the treatment of HR patients, the majority of patients received ADT. Acute and late toxicity of SBRT was not worsened by the adjunction of ADT [192]. The duration of ADT was variable going from short term (4–6 months) to long term (12–24 months). Of note, in the HYPO-RT-PC and PACE-B trials that included 89% and 91% IR disease, respectively, patients did not receive ADT. It is then difficult to be conclusive and, at present, it seems reasonable to prescribe ADT with SBRT when treating UIR and HR patients, the same way it would have been recommended if the patients had been treated with conventional EBRT. As aforementioned, the TROG 03.04 RADAR trial supports the use of long ADT versus short ADT for HR patients even with dose-escalated radiation therapy [170]. SBRT was not part of the treatment options in that trial, but the subgroup of patients treated with very high dose using a combination of EBRT and HDR-BT boost showed a significant benefit from 18 months versus 6 months ADT with a 40% relative reduction in distant progression. It seems therefore reasonable to propose the addition of ≥18 months ADT when treating HR patients with SBRT.

As previously mentioned for the FLAME randomized trial, the SIB technique can be used to deliver a focal ultra-high dose boost to a selected sub-region of the prostate where the cancer is thought to be mainly located based on biopsy mapping and mpMRI [8]. That approach can be applied to SBRT as well. In a phase II trial (Hypo-FLAME), 100 IR (25%) and HR (75%) patients were treated with SBRT in five weekly fractions to a total dose of 35 Gy on the prostate, 30 Gy to the proximal portion of the seminal vesicles and up to 50 Gy to the mpMRI-defined tumor [213]. The focal boost was conducted as high as possible as long as the organs at risk dose constraints were respected. The actual focal boost reached a median dose of 40.3 Gy. ADT was used for 62% of the patients. Preliminary results revealed a good acute safety. During the 90 days after the first radiation treatment, the cumulative incidence of grade 2 GU toxicity was 34.0%. No grade ≥3 acute GU toxicity was observed. The corresponding cumulative acute grade 2 GI toxicity rate was 5.0%. No grade 3+ acute GI toxicity was observed. In a phase I study, 55 HR patients were treated with SBRT to the prostate and the pelvis within four dose escalation cohorts [214]. The first cohort received 47.5 Gy to the prostate, 50 Gy to the mpMRI-defined intraprostatic lesion(s) and 22.5 Gy to the pelvic nodes. Radiation doses were escalated to pelvic nodes to 25 Gy and to MRI-lesion(s) to 52.5 Gy and then 55 Gy. All patients received 2 years of ADT. Dose was escalated through all four cohorts without observing any dose-limiting toxicity. With a median follow-up of 18 months, acute and late GU toxicities were 25% and 20% while acute and late GI toxicities were 13% and 7%, respectively. Late grade 3 GU and GI toxicities were 2% and 0%, respectively. However, the technicity of SBRT with simultaneous integrated ultra-high dose focal boost is very constraining and such an approach should be strictly limited to experienced facilities in clinical trials.

### 4.5. EBRT Plus SBRT Boost

Another way of achieving dose escalation for the treatment of UIR and HR disease with SBRT is to combine conventional EBRT to the prostate +/− pelvis with a SBRT boost to the prostate mimicking the combination of EBRT plus BT boost.

A retrospective study reported results of 97 HR patients treated with SBRT to the prostate at 35–36.25 Gy in five fractions (54%) or EBRT to the pelvis at 45 Gy in 25 fractions followed by a prostate SBRT boost at 19–21 Gy in three fractions (46%). ADT was given to 47% of the patients. With a 5-year median follow-up, 6-year bRFS was 69%. Overall toxicity was mild, with 5% grade 2–3 GU and 7% grade 2 GI toxicity, but the use of pelvic radiotherapy was associated with significantly higher GI toxicity (*p* = 0.001) [215].

The prospective experience with the SBRT boost approach is still weak, limited to non-randomized studies. The PROMETHEUS trial was a phase II study and included 137 IR (76%) and HR (24%) patients treated with EBRT (46 Gy/23 fractions) to the prostate +/− pelvis (11%) followed by a prostate SBRT boost (19–20 Gy/two fractions). ADT use was not specified. With a 2-year follow-up, the 2-year bRFS was 98.6% [216]. The BOOSTER trial was a phase I sequential dose escalation study and included 36 patients with IR (36%) and HR (64%) disease treated with EBRT (46 Gy/23 fractions) to the prostate + pelvis (100%) followed by a prostate SBRT boost including a SIB focal boost to the index lesion (20 Gy, 22 Gy and 24 Gy/two fractions to the prostate and 25 Gy, 27.5 Gy and 30 Gy/two fractions to the index lesion). ADT was given to 61% of patients. With a 2-year follow-up, the 3-year bRFS was 93.3% [217]. Finally, the CKNO-PRO trial included 76 IR patients treated with EBRT (46 Gy/23 fractions) to the prostate (no pelvis) followed by a prostate SBRT boost (18 Gy/three fractions). No patient received ADT. With a 5-year follow-up, the 5-year bRFS was 87.4% [218]. The tolerance reported in these trials appeared to be quite good with a cumulative incidence of acute grade 2+ GU toxicity of 24–26% (0–5% G3+), acute grade 2+ GI toxicity of 0–13% (0% G3), late grade 2+ GU toxicity of 1.4–25% (0–2% grade 3) and late grade 2+ GI toxicity of 0–9.3% (0–2% grade 3).

A single-institution retrospective propensity score-matched analysis was recently performed to compare SBRT boost with HDR-BT boost. A cohort of 232 HR patients were treated with pelvic EBRT (45 Gy/25 fractions) followed by either a SBRT boost of 19–21 Gy/two fractions (56%) or an HDR-BT boost of 19 Gy/two fractions (44%) to the prostate. Patients received ADT with a median duration of 6 months (1/3 > 18 months). The median follow-up was 73.4 months for the SBRT boost cohort and 186.0 months for the HDR-BT boost cohort. The 5- and 10-year unadjusted bRFS were 88.8% and 85.3% for SBRT boost versus 91.8% and 74.6% for HDR boost (*p* = 0.3), respectively, and the 5- and 10-year unadjusted DMFS were 91.7% and 84.3% for SBRT boost versus 95.8% and 82.0% for HDR (*p* = 0.8), respectively. After multivariate analysis, there was no significant difference in bRFS (HR 0.81; *p* = 0.6) or DMFS (HR 1.07; *p* = 0.9) between SBRT and HDR-BT boost. Similarly, after propensity score matching, there was no significant difference either in bRFS (HR 0.66, *p* = 0.4) or DMFS (HR 0.84, *p* = 0.7). Grade 3+ GU and GI toxicity in the SBRT boost cohort were 4.6% and 1.5%, versus 3.0% and 0.0% in the HDR-BT boost cohort (*p* = 0.4) [219].

Table 3 summarizes the randomized prospective trials that assess SBRT in a strategy of treatment intensification for IR and HR disease.

### 4.6. Current Recommendations for Unfavorable Intermediate Risk/High Risk Prostate Cancer

For UIR and HR disease, the addition of ADT to radiation therapy is recommended whatever technique of irradiation is used [35,140].

ABS guidelines state that BT (LDR-BT or fractionated HDR-BT) with no EBRT is an option for selected UIR disease (a single unfavorable intermediate risk factor) with organ-confined disease confirmed on mpMRI. The strength of recommendation is considered weak due to a moderate level of evidence [141,142]. Neither NCCN nor EAU guidelines recommend the use of BT with no EBRT for the treatment of UIR nor HR disease [35,140]. Still, both ABS and NCCN guidelines recognize that, in selected cases, the excellent local control provided by BT may be a sufficient therapeutic goal even in the setting of HR disease, such that BT with no EBRT can be considered as long as the goals of treatment are clearly elucidated [140,142].

ABS, NCCN and EAU guidelines consider that the association of EBRT plus BT-boost is a valid option for the treatment of UIR and HR disease. In that setting, LDR-BT, single fraction HDR-BT and fractionated HDR-BT are validated boost techniques [35,140,141]. Nevertheless, the wide application of BT boost has been limited due to controversy regarding a potential unfavorable risk-benefit balance wherein the significant increase in severe GU toxicity demonstrated in the ASCENDE-RT trial is counterbalanced by a beneficial oncologic gain of improved bRFS that has yet to be proven in terms of MFS, CSS or OS. These initial safety concerns should be reconsidered as recent technological advances as well as increasing experience with BT have improved the toxicity profile [167]. Still, it seems reasonable to consider that patients with long life expectancy and absence of significant comorbidities are the most likely to benefit from this kind of treatment intensification over the long term, by avoiding biochemical failure and subsequent systemic salvage therapy such as life-long ADT.

NCCN guidelines consider SBRT with no EBRT as an option for the treatment of UIR and HR disease for selected patients when longer courses of EBRT would present medical or social hardship [140]. EAU guidelines recommend restricting the use of SBRT to prospective clinical trials and to inform patients on the uncertainties of the long-term outcome [35]. SBRT with EBRT is not recommended outside clinical trials.

## 5. Conclusions

SBRT is a validated option for the treatment of LR and FIR prostate cancer knowing that active surveillance should be favored. Yet, BT remains a strong competitor, especially for young patients, as series with 10–15 years of median follow-up confirmed its efficiency over time. SBRT has the advantage of less acute urinary toxicity. Its superiority with regard to reduced sexual impairment has yet to be proven. For UIR and HR disease, BT can be proposed as a prostatic boost in combination with pelvic EBRT provided that patients are informed of an increased risk of severe urinary toxicity. For this set of patients, prostate SBRT is an option if the risk of pelvic lymph node involvement is considered low enough (<20%). Otherwise, prostate SBRT is not validated even if it can be considered for older patients. Treatment intensification strategies such as elective pelvic nodal ultra-hypofractionated irradiation, combination of EBRT plus SBRT prostatic boost or SBRT with focal ultra-high boost to the index lesion should be considered investigational. Magnetic resonance-guided SBRT could further improve outcomes of SBRT, but more numerous and mature results are needed to justify the inherent cost implications of the technique. SBRT remains under active evaluation with several randomized trials pending. If the technique reinforces its promising results in a longer follow-up, it may then demonstrate its superiority against BT in the near future.

## Figures and Tables

**Table 1 cancers-14-02226-t001:** Non-randomized prospective series assessing, with a very long follow-up (>5 years), biochemical outcome of localized prostate cancer treated with brachytherapy (BT).

	NCCN Risk Groups (n)			bRFS (%) [Time Point]	
Authors	LR	FIR	FU (y)	LR	FIR
Sylvester 2011 [44]	128	36	11.7	86 [15y]	80 [15y]
Morris 2013 [45]	586	419	7.5	94 [10y]	
Kittel 2015 [46]	1219	592	6.8	87 [10y]	79 [10y]
Cosset 2016 [47]	452	223	11.0	87 [10y]	71 [10y]
Wilson 2016 [48]	90	84	7.8	96 [10y]	91 [10y]
Langley 2017 [49]	316	220	8.9	95 [10y]	90 [10y]
Prada 2018 [50]	229	41	9.2	94 [15y]	76 [15y]
Jacobsen 2018 [51]	206	265	6.6	90 [10y]	75 [10y]
Winoker 2019 [52]	241	89	9.9	93 [15y]	83 [15y]
Vuolukka 2019 [53]	142	85	11.4	85 [10y]	72 [10y]
Lazarev 2019 [54]	370	170	12.5	86 [17y]	80 [17y]

LR = low risk; FIR = favorable intermediate risk; FU = median follow-up; bRFS = biochemical recurrence-free survival.

**Table 2 cancers-14-02226-t002:** Non-randomized prospective series (n > 40) assessing biological outcome of localized prostate cancer treated with stereotactic body radiation therapy (SBRT).

		NCCN Risk Groups (%)						bRFS (%)	
Authors	n	LR	IR	HR	Gy/fx	ADT (%)	FU (y)	2–3y	5y
Madsen 2007 [73]	40	100	0	0	33.5/5	NR	3.4	90.0	
Friedland 2009 [74]	112	NR	NR	NR	35–36/5	19	2.0	97.4	
Kang 2011 [75]	44	11	23	66	32–36/4	89	3.3	100 ^a^ 96.0 ^b^	100 ^a^ 90.9 ^b^
Mc Bride 2012 [76]	45	100	0	0	36.25–37.5/5	0	3.7	97.7	
King 2012 [77]	67	100	0	0	36.25/5	0	2.7	100	94.0
Aluwini 2013 [78]	50	60	40	0	38/4	0	1.9	100	
Chen 2013 [79]	100	37	55	8	35–36.25/5	11	2.3	99.0	
Bolzicco 2013 [80]	100	41	42	17	35/5	29	3.0	96.0	94.4
Loblaw 2013 [81]	84	100	0	0	35/5	0	4.6	100	98.0
Oliai 2013 [82]	70	51	31	17	35–37.5/5	33	2.6	94.5	
Lee 2014 [83]	45	13	58	29	36/5	38	5.2	~95.0	89.7
Mantz 2014 [84]	102	100	0	0	40/5	NR	5.0	100	100
Fuller 2014 [27]	79	51	49	0	38/4	NR	5.0	100	95
Bernetich 2014 [85]	142	43	44	13	35–37.5/5	28	3.3	95.5	92.7
Davis 2015 [86]	437	43	49	8	35–38/4–5	11	1.7	96.1	
Freeman 2015 [87]	1743	41	42	10	35–40/4–5	NR	2.0	92.0	
Rana 2015 [88]	102	36	55	8	36.25/5	9	4.3	100	
Hannan 2016 [89]	91	36	64	0	45–50/5	17	4.5	100	98.6
D’Agostino 2016 [90]	90	59	41	0	35/5	13	2.3	97.8	
Rucinska 2016 [91]	68	10	90	0	33.5/5	77	2.0	100	
Katz 2016 [92]	515	63	30	7	35–36.25/5	14	7.0	98.0 ^a^ 72.0 ^b^	94.7 ^a^ 68.6 ^b^
Dixit 2017 [93]	45	24	62	13	36.25/5	16	1.5	100	
Miszczyk 2017 [94]	400	53	47	0	36.25/5	58	1.3	99.5	
Koskela 2017 [95]	218	22	27	51	35–36/5	65	1.9	~97.0 ^a^ 92.8 ^b^	
Jackson 2018 [96]	66	49	51	0	37/5	0	3.0	100	
Alayed 2018 [97]	84 30	100 60	0 40	0 0	35/5 40/5	1 0	9.6 6.9	100 100	97.5 96.7
Meier 2018 [98]	309	172	137	0	40/5	0	5.1		97.1

LR = low risk; IR = intermediate risk; HR = high risk; Gy = prescription dose in Gray, fx = number of fractions; bRFS = biochemical recurrence-free survival; ADT = androgen deprivation therapy; FU = median follow-up. When a significant subpopulation (n ≥ 30) of HR patients was available, results were split between LR/IR ^a^ and HR ^b^.

**Table 3 cancers-14-02226-t003:** Randomized prospective trials assessing stereotactic body radiation therapy (SBRT) as a treatment for localized prostate cancer.

Trial Identifier	Coutnry	Risk Groups	Arms (Standard/Experimental)	Primary Outcome	Measure
**NCT03830788** *TEMPOS*	France	LR and IR	LDR-BT SBRT	Medico-economic	Cost-utility analysis
**NCT02895854** *BRAVEROBO*	Finland	LR and IR	LDR-BT SBRT	Toxicity	CTCAE
**NCT04870567**	Russia	LR and IR	HDR-BT SBRT	Toxicity and erectile dysfunction	CTCAE and PROs
**NCT03525262** *POTEN-C*	USA	LR and IR	SBRT SBRT (neurovascular sparing)	Sexual toxicity	PROs (Sexual function)
**NCT05019846** *SPA*	Italy	IR and HR	SBRT SBRT + 6m ADT	Efficacy	bRFS
**NCT03056638**	USA	IR	SBRT SBRT + 6m ADT:	Efficacy	2y biopsies
**NCT03253978** *SPORT*	UK	HR	SBRT + ADT SBRT with pelvis (SIB) + ADT	Toxicity	CTCAE and PROs
**ISRCTN45905321** *HYPO-RT-PC*	Sweden	IR and HR	EBRT SBRT	Efficacy	bRFS
**NCT01584258** *PACE A B C*	UK	LR and IR LR and IR IR and HR	*PACE-A* Prostatectomy SBRT *PACE-B* EBRT SBRT *PACE-C* EBRT + ADT SBRT + ADT	Efficacy Efficacy Efficacy	bRFS bRFS bRFS
**NCT01794403** *HEAT*	USA	LR and IR	EBRT SBRT	Efficacy	bRFS and 2y biopsies
**NCT03367702** *NRG-GU005*	USA	IR	EBRT SBRT	Toxicity	PROs
**NCT02361515** *RPAH2*	France	LR and IR	EBRT SBRT (rectal spacer)	Toxicity	CTCAE
**NCT02594072** *ASSERT*	Canada	IR and HR	EBRT + ADT SBRT + ADT	Toxicity	CTCAE
**NCT03380806** *PBS*	Canada	HR	EBRT + ADT EBRT + SBRT boost + ADT	Toxicity	PROs
**NCT01839994**	Poland	IR and HR	EBRT + ADT EBRT + HDR-BT boost + ADT or EBRT + SBRT boost + ADT	Efficacy	bRFS
**NCT02300389** *HYPOPROST*	Poland	HR	EBRT + ADT EBRT + SBRT boost + ADT	Efficacy	bRFS
**NCT04100174**	Canada	IR and HR	EBRT + HDR-BT boost SBRT + focal HDR-BT boost	Urinary toxicity	PROs (Urinary function)

LR = low-risk; IR = intermediate-risk; HR = high-risk; LDR = low-dose rate; HDR = high-dose rate; BT = brachytherapy; SBRT = stereotactic body radiation therapy; EBRT = external beam radiation therapy; ADT = androgen deprivation therapy; SIB = simultaneous integrated boost; bRFS = biological recurrence free survival; CTCAE = Common Terminology Criteria for Adverse Events; PROs = patient reported outcomes.

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
