# Peer review of "Stereotactic Radiation Therapy versus Brachytherapy: Relative Strengths of Two Highly Efficient Options for the Treatment of Localized Prostate Cancer"

_cancers, 2022, doi:10.3390/cancers14092226_

Round 1
Reviewer 1 Report
- General comment (originality, scientific accuracy, strengths and/or weaknesses):
The manuscript entitled “Stereotactic Radiation Therapy versus Brachytherapy. Relative Strengths of Two Highly Efficient Option for the Treatment of Localized Prostate Cancer” is a large review aiming to report the current levels of evidence of SBRT and BT for the treatment of localized prostate cancer. The manuscript is well written and quite comprehensive of the available literature. In addition, I really appreciated your work which, although extensive, is readable and fluent. Few corrections are required, in addition to minor issues such as typos and lack of references. Probably I would also revise the headings of each section in order to avoid the “wall of text” effect and provide an easier reading also for non-expert readers.
- Major correction
INTRODUCTION
38-43: I appreciate a precise and concise introduction, but in this case, it is too straightforward. Begin the article with some epidemiological data or historical utilization of radiation therapy in prostate cancer.
52-61: This is much better for example
63: Be more precise on the potential reasons which brought to the loss of appealing of BT
RADIOBIOLOGY
72: I would break the paragraph into two subparagraphs regarding EBRT and BT
73-78: I do not understand why starting this paragraph comparing EBRT and DE-EBRT. I think it would more suitable to move this part after reporting the radiosensitivity of prostate cancer.
222: This is a complicated topic that should be better reported and explained
515: Report how erectile function was assessed in the studies evaluated
ULTRA-HYPOFRACTIONATED RADIOTHERAPY VERSUS BRACHYTHERAPY
602: I would utilize a more homogeneous reporting, similar to the previous paragraph, in terms of subparagraphs and tables
PATENTS?
TABLES
Did you consider the possibility to use comprehensive tables as for the radiobiology paragraph?
- Minor corrections
RADIOBIOLOGY
129: Write in full the 4Rs
242-249: Add references
468-477: Add references
550: check typos
586: check typos
588-596: add references
TABLES
Table 1: define very long follow up
Author Response
General comment (originality, scientific accuracy, strengths and/or weaknesses):
The manuscript entitled “Stereotactic Radiation Therapy versus Brachytherapy. Relative Strengths of Two Highly Efficient Option for the Treatment of Localized Prostate Cancer” is a large review aiming to report the current levels of evidence of SBRT and BT for the treatment of localized prostate cancer. The manuscript is well written and quite comprehensive of the available literature. In addition, I really appreciated your work which, although extensive, is readable and fluent. Few corrections are required, in addition to minor issues such as typos and lack of references. Probably I would also revise the headings of each section in order to avoid the “wall of text” effect and provide an easier reading also for non-expert readers.
Response: The headings of the main sections have been shortened.
Major correction
INTRODUCTION
38-43: I appreciate a precise and concise introduction, but in this case, it is too straightforward. Begin the article with some epidemiological data or historical utilization of radiation therapy in prostate cancer.
Response: The “Introduction” section has been reworded in order to better highlight the history of prostate BT development and the history of EBRT technological improvements that led to treatment intensification strategies such as DE-EBRT and hypofractionation.
52-61: This is much better for example
63: Be more precise on the potential reasons which brought to the loss of appealing of BT
Response: In the “Introduction” section, we listed some of the factors that may have reduced the attractivity of BT.
RADIOBIOLOGY
72: I would break the paragraph into two subparagraphs regarding EBRT and BT
Response: We divided the paragraph into two subparagraphs for SBRT and BT
73-78: I do not understand why starting this paragraph comparing EBRT and DE-EBRT. I think it would more suitable to move this part after reporting the radiosensitivity of prostate cancer.
Response: The text referring to DE-EBRT has been moved from the “Radiobiology” section and placed in the “Introduction” section to better illustrate the timeline of strategies for EBRT intensification.
222: This is a complicated topic that should be better reported and explained
Response: We agree that addressing the fact that young people are often referred to surgery rather than brachytherapy was somewhat subjective and controversial. This observation is outside of the main scope of the current paper. We therefore decided to remove that comment and to only underscore that BT should be proposed as a treatment option to young patients.
515: Report how erectile function was assessed in the studies evaluated
Response: We indicated what patient-reported outcome measurement scales were used in the different studies our paper is referring to.
ULTRA-HYPOFRACTIONATED RADIOTHERAPY VERSUS BRACHYTHERAPY
602: I would utilize a more homogeneous reporting, similar to the previous paragraph, in terms of subparagraphs and tables
Response: We understand the benefit of a homogeneous presentation, but we could not replicate the same subparagraphs in section 4 as those used in section 3 for the following reasons:
When looking at low-risk and favorable intermediate-risk prostate cancers, BT and SBRT are two options that are proposed as single treatments. The successive description of oncologic outcomes followed by toxicity outcomes for these two options is readable.
On the other hand, in the field of unfavorable intermediate-risk and high-risk prostate cancers, BT and SBRT have been tested as single treatments but have also been tested in several strategies of treatment intensification including combination of EBRT plus BT boost, EBRT plus SBRT boost, SBRT with ultra-focal dose escalation on the index lesion, SBRT with pelvic irradiation... For section 4, we wanted to look at each strategy in a cohesive way presenting their respective oncologic and toxicity outcomes together.
PATENTS?
Response: There is indeed no reason to have a section named patents. It has been removed.
TABLES
Did you consider the possibility to use comprehensive tables as for the radiobiology paragraph?
Response: We used comprehensive tables in section 3 because the numerous publications we referenced had similar designs and objectives (non-randomized prospective cohorts assessing oncologic outcomes of BT or SBRT used as single treatments) making it possible to synthesize data in tables.
In section 4, references are very heterogeneous in terms of study designs (retrospective studies, non-randomized prospective cohorts, randomized trials with very preliminary or mature results, comprehensive literature reviews, national cancer database analysis, propensity score matched analyses…). These studies are also very heterogeneous in terms of objectives and questions asked due to the fact that SBRT and BT were tested in various strategies of treatment intensification. It was therefore not possible to summarize these heterogeneous data exhaustively in a single table.
- Minor corrections
RADIOBIOLOGY
129: Write in full the 4Rs
Response: Done
242-249: Add references
Response: We added reference [58] that reports results of the 44/20 trial. To our knowledge, the 20/0 trial results have been reported only in the pooled analysis already referenced in the manuscript.
[58] Merrick, G.S.; Wallner, K.E.; Butler, W.M.; Galbreath, R.W.; Taira, A.V.; Orio, P.; Adamovich, E. 20 Gy versus 44 Gy of Supplemental External Beam Radiotherapy with Palladium-103 for Patients with Greater Risk Disease: Results of a Prospective Randomized Trial. Int J Radiat Oncol Biol Phys 2012, 82, e449-455, doi:10.1016/j.ijrobp.2011.07.016.
468-477: Add references
Response: We added reference [107] that provides full description of the technique used for SBRT in the HYPO-RT-PC trial.
[107] Widmark, A.; Gunnlaugsson, A.; Beckman, L.; Thellenberg-Karlsson, C.; Hoyer, M.; Lagerlund, M.; Kindblom, J.; Ginman, C.; Johansson, B.; Björnlinger, K.; et al. Ultra-Hypofractionated versus Conventionally Fractionated Radiotherapy for Prostate Cancer: 5-Year Outcomes of the HYPO-RT-PC Randomised, Non-Inferiority, Phase 3 Trial. The Lancet 2019, 394, 385–395, doi:10.1016/S0140-6736(19)31131-6.
550: check typos
Response: Indentified has been replaced by Identified
586: check typos
Response: We put a missing dot at the end of the sentence.
588-596: add references
Response: All the statements in that section are part of the ABS guidelines that were already referenced (ref [145]).
[145] Low Dose Rate Brachytherapy for Primary Treatment of Localized Prostate Cancer: A Systemic Review and Executive Summary of an Evidence-Based Consensus Statement.
TABLES
Table 1: define very long follow up
Response: We completed the title using “with a very long follow-up (>5 years)”
Reviewer 2 Report
This review of the literature gives a great overview of the relative merits of SBRT versus brachytherapy for localised prostate cancer.
The structure of the paper is logical. Its message clear, complete and concise.
It is hard to find suggestions, since discussion of the literature is complete and supports the message of the paper.
Perhaps the fractionation in HDR-BT and the relation with better BRFS could be seen as an indication of the success of fractionated SBRT (lines 267-269), which could be noted in the paper instead of only describing the difference in the trial. It could be illustrative to also add the HDR-BT results in table 2, but this is optional.
But that is if I have to make a remark about the paper at all.
Also, the SPIRIT trial is mentioned a couple of times. Maybe add the poor accrual of this trial, making it difficult to base solid conclusion on this study.
Thanks for the informative and great read!
Some minor spelling errors were encountered:
Line 49: worldwide (worldilde).
Line 80: range (ranging).
Line 359: pollakisuria (pollakiuria).
Line 1050: recommendations (recommandations).
Author Response
This review of the literature gives a great overview of the relative merits of SBRT versus brachytherapy for localised prostate cancer.
The structure of the paper is logical. Its message clear, complete and concise.
It is hard to find suggestions, since discussion of the literature is complete and supports the message of the paper.
Perhaps the fractionation in HDR-BT and the relation with better BRFS could be seen as an indication of the success of fractionated SBRT (lines 267-269), which could be noted in the paper instead of only describing the difference in the trial.
Response: We added the following sentence to the manuscript:
Single ultra-high dose schemes should be viewed with caution as they were tested inferior to fractionated schemes with HDR-BT.
It could be illustrative to also add the HDR-BT results in table 2, but this is optional.
Response: Table 2 is already quite heavy and we hope you will not mind that we decided to limit it to the results of SBRT without adding HDR-BT.
But that is if I have to make a remark about the paper at all.
Also, the SPIRIT trial is mentioned a couple of times. Maybe add the poor accrual of this trial, making it difficult to base solid conclusion on this study.
Response: We added the following comment:
The SPIRIT trial suffered from poor accrual making it difficult to derive a solid conclusion from this study, but with a median follow-up of 5 years, the ability to have an erection was preserved after BT for more than 60% of patients and was sufficient for intercourse in more than 50% of cases (sexual domain of the cancer–specific Expanded Prostate Cancer Index Composite – EPIC-26).
Thanks for the informative and great read!
Some minor spelling errors were encountered:
Line 49: worldwide (worldilde).
Response: Modified
Line 80: range (ranging).
Response: We used “it is now consensus that a total dose ranging between 74 Gy and 80 Gy (conventional fractionation of 1.8-2 Gy per fraction) is recommended”
Line 359: pollakisuria (pollakiuria).
Response: Modified
Line 1050: recommendations (recommandations).
Response: Modified